# Occurrence and Risk Factors for Macular Edema in Patients with Juvenile Idiopathic Arthritis-Associated Uveitis

**DOI:** 10.3390/jcm10194513

**Published:** 2021-09-29

**Authors:** Christoph Tappeiner, Han Sol Bae, Kai Rothaus, Karoline Walscheid, Arnd Heiligenhaus

**Affiliations:** 1Department of Ophthalmology, University Hospital Essen, University of Duisburg-Essen, 45141 Essen, Germany; christoph.tappeiner@pallas-kliniken.ch; 2Department of Ophthalmology, Pallas Kliniken, 4600 Olten, Switzerland; 3Department of Ophthalmology, Inselspital, Bern University Hospital, University of Bern, 3100 Bern, Switzerland; 4Department of Ophthalmology, St. Franziskus Hospital, 48145 Muenster, Germany; hsbae1129@gmail.com (H.S.B.); kai.rothaus@augen-franziskus.de (K.R.); karoline.walscheid@uveitis-zentrum.de (K.W.); 5University of Duisburg-Essen, 45141 Essen, Germany

**Keywords:** juvenile idiopathic arthritis, uveitis, macular edema, risk factors, occurrence

## Abstract

Purpose: To analyze occurrence and risk factors for macular edema (ME) in juvenile idiopathic arthritis-associated uveitis (JIA-U). Methods: Retrospective analysis of patients with JIA-U at a tertiary referral uveitis center between 2000 and 2019. Epidemiological data and clinical findings before ME onset were evaluated. Results: Out of 245 patients, ME developed in 41 (18%) of the 228 JIA-U patients for whom data documentation was complete during the follow-up (mean 4.0 ± 3.8 years). Risk factors (univariable logistic regression analysis) at baseline for subsequent ME onset included older age at initial documentation at institution (hazard ratio, HR 1.19, *p* < 0.0001), longer duration of uveitis at initial documentation (HR 1.17, *p* < 0.0001), worse best-corrected visual acuity (BCVA; HR 2.49, *p* < 0.0001), lower intraocular pressure (IOP; HR 0.88, *p* < 0.01), band keratopathy (HR 2.29, *p* < 0.01), posterior synechiae (HR 2.55, *p* < 0.01), epiretinal membrane formation (HR 6.19, *p* < 0.0001), optic disc swelling (HR 2.81, *p* < 0.01), and cataract (HR 4.24, *p* < 0.0001). Older age at initial documentation at institution (HR 1.55, *p* < 0.001), worse BCVA (HR 28.56, *p* < 0.001), and higher laser-flare photometry (LFM) values (HR 1.003, *p* = 0.01) were independent risk factors for ME manifestation. Patients with ME revealed significant changes in BCVA, LFM, and IOP and new optic disc swelling at 6 and 3 months before ME onset compared to timepoint of ME occurrence (*p* < 0.05, each). Conclusion: ME is a common complication of JIA-U. Demographic risk factors and courses of IOP, BCVA, and LFM may indicate patients at risk for ME onset.

## 1. Introduction

Juvenile idiopathic arthritis (JIA) comprises a group of rheumatoid diseases with onset before 16 years of age [1,2]. In 9–25% of children, uveitis manifests during the course of disease [3,4,5,6,7]. As uveitis may lead to potentially irreversible vision-threatening complications [4,8,9,10,11,12], identifying patients at risk for such complications is crucial for improving final disease outcome.

Macular involvement has been reported in up to 82% of patients with JIA-U when screening with highly sensitive optical coherence tomography (OCT) [13]. Different JIA-U studies have reported that macular edema (ME) becomes manifest in JIA-U [3,4,10,14,15,16]. Importantly, if children are followed up long enough, ME rates of up to 30–35% have been observed [17,18]. ME may cause legal blindness in up to 8% of children with uveitis [19], especially if there are delays in diagnosing uveitis and in achieving ocular quiescence [14,20,21]. In a healthy eye, osmotic factors, hydrostatic forces, capillary permeability, and tissue compliance normally prevent extracellular intraretinal fluid and proteins from accumulating [21,22]. In uveitic eyes, however, the blood–retina barriers might break down, for example, due to the release of proinflammatory cytokines [23,24]. Furthermore, ME may also be triggered after intraocular surgery in such eyes [25]. 

The aim of this study was to analyze the occurrence and risk factors for ME in a large cohort of patients with JIA-U. 

## 2. Methods

All children with JIA-U followed up between 30 June 2000 and 20 March 2019 at the tertiary uveitis center of the Department of Ophthalmology at the St. Franziskus-Hospital, Muenster, Germany, were included in this retrospective study. This study was performed in accordance with the Declaration of Helsinki and after obtaining the approval of the responsible ethics committee (2012-403-f-S). 

JIA was defined according to the International League of Associations for Rheumatology (ILAR) [26], whereas uveitis was classified according to the Standardized Uveitis Nomenclature [27]. Data were collected using a standardized protocol, documenting medical history, laboratory results, rheumatological findings, laser-flare photometry (LFM) values (FM 500, Kowa, Tokyo, Japan), best-corrected visual acuity (BCVA; Lea or E-tests for children < 7 years and Snellen for adolescents), OCT measurements (Spectralis HRA-OCT, Heidelberg Engineering, Heidelberg, Germany), tonometry values (applanation tonometer, Haag Streit, Koeniz, Switzerland; or i-Care tonometer, Icare, Vantaa, Finland), slit-lamp appearance, and fundoscopy findings typical for JIA-U. The diagnosis of ME was based on fundoscopy, OCT and/or fluorescein angiography findings. At each visit, patients were observed independently by two ophthalmologists. 

The initial visit at the uveitis center was defined as baseline. One eye per patient was analyzed for this study: in unilateral uveitis, the uveitic eye was selected; for bilateral uveitis, the eye with the higher anterior chamber cell (ACC) grade at initial presentation was analyzed; and in bilateral uveitis with identical ACC grades, the eye with higher LFM values, presence of posterior synechiae, and/or more severe uveitic complications was selected. For patients in whom ME became manifest during follow-up, the timepoints 3 and 6 months before ME onset were evaluated to identify predicting factors of ME. The presence of typical uveitic complications such as band keratopathy, synechiae, iris rubeosis, cataract, ME, epiretinal membrane, ocular hypertension (IOP ≥ 21 mmHg), glaucoma (pathological optic disc and/or visual fields), ocular hypotonia (IOP ≤ 6 mmHg), and optic disc swelling was documented using a standardized form at baseline and for eyes with subsequent ME onset 6 and 3 months before and at time of ME occurrence, too. Topical and systemic anti-inflammatory treatments at baseline and at each follow-up visit were documented. As this was a retrospective study, treatments were adjusted at the discretion of the treating ophthalmologists and/or pediatric rheumatologists. 

Statistical analysis was performed with MedCalc Statistical Software (version 20.009, MedCalc Software, Ostend, Belgium) and R (version 4.0.2, R Foundation). Normal distribution was tested by Shapiro–Wilk normality test. T-tests and ANOVA tests were used for group comparison, as appropriate. Otherwise, Wilcoxon tests and Kruskal–Wallis tests were applied as a nonparametric alternative. For nominally scaled variables, Chi² test, or alternatively, Fisher’s exact test were performed.

Risk factors for ME occurrence were evaluated by univariable Cox survival regression analysis and multivariable Cox survival regression analysis to calculate the HR. Thereby, we initialized the model with promising factors from the univariable analysis and, in a stepwise fashion, further reduced the model to converge sufficiently. *p* values less than 0.05 were defined as statistically significant. 

## 3. Results

Out of 245 patients with JIA-U, 228 patients for whom data documentation was complete were included in this study. The epidemiological data are summarized in Table 1. The mean age at JIA onset and uveitis manifestation were 4.0 ± 3.0 and 5.2 ± 3.1 years, respectively. The mean age at baseline visits at our uveitis center was 7.6 ± 4.3 years. The mean observation period for the ME occurrence analysis was 6.8 ± 4.2 years (range 0.0 to 18.8), and the mean follow-up with ME was 4.0 ± 3.8 years. Uveitis was bilateral in 72.8% of children. Most patients (68.9%) were girls, ANA positive (94.2%), and presented with oligoarthritis (75.9%) or polyarthritis (17.5%) JIA subtypes.

In total, ME was present in 41 (18.0%) study eyes during the entire observation period (Figure 1). At baseline, ME was found in ten study eyes (4.4%), whereas it developed during further follow-up in an additional 31 eyes (13.6%) after a mean uveitis duration of 5.3 ± 3.5 years. Patients in whom ME developed were significantly more often of female sex (*p* < 0.05) and older at baseline (*p* < 0.001) and presented with a longer duration of uveitis at baseline (*p* = 0.001) than patients without ME, whereas JIA subtype and ANA positivity were comparable between the two groups (Table 1). Risk factors for ME onset in patients without prior ocular surgery are shown in Table 2. The univariable regression analysis identified older age at initial documentation at institution (hazard risk HR 1.19, *p* < 0.0001), longer duration of uveitis at initial documentation (HR 1.17, *p* < 0.0001), worse BCVA (HR 2.49, *p* < 0.001), lower IOP (HR 0.88, *p* < 0.01), presence of band keratopathy (HR 2.29, *p* < 0.01), posterior synechiae (HR 2.55, *p* < 0.01), epiretinal membrane (HR 6.19, *p* < 0.0001), optic disc swelling (HR 2.81, *p* = 0.02), and cataract (HR 4.24, *p* < 0.0001) as risk factors for ME onset. Older age at initial documentation at institution (HR 1.55, *p* < 0.001), worse BCVA (HR 28.56, *p* < 0.001), and higher LFM (HR 1.003, *p* = 0.01) were independent risk factors for ME manifestation in a multivariable regression analysis. The systemic treatment at timepoint of ME onset and a potential temporal association of ME with a previous intraocular surgery are shown in Figure 1. At timepoint of ME occurrence, patients were treated with adalimumab (*n* = 16), infliximab (*n* = 2), abatacept (*n* = 1) and golimumab (*n* = 1).

In Table 3, ocular findings 6 and 3 months before and at timepoint of ME onset are shown. A significant change in BCVA, LFM, IOP, and new optic disc swelling were observed at timepoint of ME onset compared to 3 and 6 months before (*p* < 0.05, each). In these data, no significant change in topical or systemic treatment was found within 6 months before ME onset. Compared to 3 months before, the percentage of patients with bDMARDs at timepoint of ME onset slightly but not significantly decreased (Table 3), as bDMARDs had to be stopped due to adverse events or interruption of health insurance cost approval. The course of laser flare, BCVA, IOP, and central foveal thickness before and at ME onset is illustrated in Figure 2. LFM values increased significantly, whereas BCVA and IOP decreased significantly in the 6 months before ME onset. 

## 4. Discussion

In patients with JIA-U, harmful and potentially sight-threatening complications can develop. Even in the era of biological DMARDs, the disease still bears the risk of irreversible visual impairment [28,29]. Various studies have analyzed ocular complications in JIA-U, ME being one of them [30,31,32,33]. In recent years, international consensus papers and randomized trials for the management of JIA-U have been published, aiming to reduce burden of disease [34,35,36]. Furthermore, diverse outcome measures have been proposed by the Multinational Interdisciplinary Working Group for Uveitis in Childhood (MIWGUC), also including typical structural complications such as ME [37]. 

In our JIA-U cohort, ME was observed in 18% of patients during a follow-up period of 6.8 years. A high variability in ME occurrence rates has been reported in the JIA-U studies conducted in recent decades. This variability is due to different inclusion criteria of patients, different screening approaches for detecting ME, changes in treatment strategies over the decades, particularly as a result of innovative anti-inflammatory drugs for managing JIA-U (such as bDMARDs), and different study designs, including retrospective single-center or multi-center case series and prospective population-based registries. Consequently, ME rates reported for JIA-U ranged from 5.6% and up to nearly half of the cases [6,13,32,33,38,39,40]. The findings within our present study from a tertiary referral center are in the mid-range of the occurrence rates reported in previous publications. 

In our study, ME was already present in 10 out of 228 of our JIA patients at baseline. In another 31 patients, ME onset was observed after a mean uveitis duration of 5.3 ± 3.5 years (range 0.2–13.1 years). This rate is in line with previous publications, which have shown that ME may develop at intermediate and later stages of JIA-U [14], and particularly in a severe inflammatory uveitis disease course. However, ME was occasionally observed already at presentation [41], which is not unusual for JIA-U and is characteristic for an insidious onset of flare even in eyes with high uveitis activity. 

In the current study, we analyzed risk factors at baseline for subsequent ME onset. Older age at first presentation, worse BCVA, and high LFM values at baseline were identified as independent risk factors in a multivariable analysis, whereas other parameters, such as uveitis duration at baseline, lower IOP, and presence of other uveitis-related complications at baseline (band keratopathy, posterior synechiae, epiretinal membrane, optic disc swelling, and cataract), were found to be risk factors in a univariable regression analysis. Presence of secondary complications at baseline indicates that a patient had previously experienced a severe course of uveitis in the affected eyes, most probably with high inflammatory activity. It may be assumed that such an inflammatory activity with the release of proinflammatory cytokines in the long-term causes the blood–aqueous barrier to break down. This notion is also in line with the findings from a study of Paroli et al., who reported that eyes with severe ocular complications at first visit ended with an unfavorable visual prognosis [16], which correlates well with worse BCVA in our study. This highlights the necessity for an early diagnosis in order to initiate appropriate therapy and to potentially reduce the morbidity from disease. In a previous study, de Boer et al. [14] also found a correlation between duration of uveitis and the risk for ME. In our study, however, demographic (age at JIA or uveitis onset, sex, or JIA subtype) and laboratory factors (e.g., ANA positivity), though important for JIA-U disease occurrence, did not indicate individual risk for ME onset. 

Our observation again highlights the clinical value of LFM for monitoring JIA-U. Not only were higher baseline values a risk factor for subsequent ME onset (Table 1 and Table 2), but we also observed a significant increase in LFM values in the 6 months before ME onset (Figure 2). An increase in LFM indicates a blood–aqueous barrier dysfunction due to uveitis activity. Previously, our group found a correlation between LFM and ocular hypotony in JIA-U [42]. The present study indicates that LFM is also relevant as a risk factor for ME. Monitoring LFM is a useful diagnostic tool and may be more sensitive for measuring barrier dysfunction than only evaluating ACC in JIA-U patients [12].

It is well known that optic disc swelling in JIA-U is a relevant accompanist of intraocular inflammatory activity. Optic disc swelling at baseline and during follow-up also indicated a risk for subsequent ME manifestation (Table 1, Table 2 and Table 3). 

In our patients, a significant decrease in BCVA was observed at ME onset. It is known that ME is a relevant factor for burden of disease in JIA-U. Visual impairment due to uveitic complications in JIA-U has been reported in 18–38% of patients [11,32,33,43,44,45]. Notably, ME occurs in 3 to 47% of patients and persisting ME leads to visual impairment and potentially also to legal blindness in up to 8% of these children [9,14,16,24,46]. However, secondary complications at initial presentation (2%) and during follow-up (14%; 0.03/eye-year) were not significantly associated with visual acuity loss in a study by Paroli et al. [47]. In a retrospective study, de Boer et al. evaluated 123 patients with childhood uveitis, 25 of them with JIA-U, during an average follow-up of 3 years [19]. They found that in 8% of patients with visual acuity < 20/200 this was due to ME. In another retrospective study of 65 patients (117 eyes) with 5 years of follow-up, a total of 12 blind eyes in 10 patients were identified and ME was detected in 3 of them (25%) [48]. In line with these notions, ME was not only associated with worse visual acuity but also with a decrease in contrast sensitivity in another study [49], which highlights the fact that ME is a relevant factor for impairment of visual function in JIA-U. In our study, too, worse BCVA at initial presentation was associated with a higher rate of ME onset during the further course of disease. Presumably, eyes with low visual acuity at baseline were already severely affected from chronic inflammation, and therefore were at higher risk for further complications. Although all our patients were managed according to international treatment recommendations using a step-ladder approach from topical corticosteroids, to csDMARDs, and to bDMARds, ME onset could still not be prevented in patients at risk. 

In our experience, OCT measurements are feasible and very useful for monitoring children with JIA-U. By using OCT, the macula can be noninvasively imaged to sensitively detect ME, even when it cannot be visualized by ophthalmoscopy. In an OCT study of patients with JIA-U, maculopathy was found in 84% of 61 eyes (perifoveal thickening 74%, ME 48%, foveal detachment 18%, and atrophic changes 10%) with a Zeiss OCT 3 device [13]. Using sensitive diagnostics including OCT imaging, posterior complications in JIA-U were reported in 1/3 of patients in another study from Italy [16]. Within this study, ME was found in 13.8% clinically and in 25% by using OCT imaging, highlighting that such complications are underestimated by clinical examination. Consequently, OCT has been established as a routine tool for taking care of uveitic patients in recent decades [13].

Strong evidence for the beneficial effect of cs and bDMARDs, especially adalimumab, on the disease course of JIA-U was previously published by our group based on data from a prospective nationwide database in Germany [3,50]. However, in the current study, DMARD treatment at baseline did not correlate with the risk for the subsequent occurrence of ME (Table 2). Furthermore, no significant change in topical and/or systemic treatment was observed in the 6 months before ME onset (Table 3). We assume that the present study was underpowered to analyze the impact of DMARD treatment on JIA-U onset. We can only speculate whether adjusting treatment in the months before ME onset—if early predictors would have been recognized—might have prevented ME from developing in our patients. The present study results will help to identify such predictors for the management of JIA-U patients in the future. 

In the current study, ocular surgery did not play a relevant role as a risk factor for ME onset. Indeed, most of the patients in whom ME developed had not undergone surgery in the previous 3 months (Figure 1, Table 1), and surgery prior to baseline was not a risk factor for ME (Table 1). However, intraocular surgery may lead to ME in uveitis patients, as the surgical trauma can disrupt the eye barriers and induce an additional release of proinflammatory cytokines. Therefore, proper control of uveitis is crucial perioperatively. However, ME can develop even when uveitis is well controlled for months or even years after surgery [51]. Indeed, ME was reported in 16 of 25 JIA-U patients (with visual impairment <0.3 in 5 eyes) after cataract surgery, despite the fact that DMARDs and topical corticosteroid treatment were carefully adjusted [25]. The authors pointed out that, ideally, uveitis should be completely quiescent before performing surgery. Indeed, there have been reports of lower or no significant risk for developing ME, ocular hypertension, secondary glaucoma, or disc swelling after ocular surgery even in a long-term follow-up of 10 years when maximum control of perioperative inflammation and intensive follow-up is achieved [15]. Our group also reported a favorable outcome without onset of ME after cataract surgery and intraocular lens implantation in JIA-U in a retrospective analysis of 16 JIA-U patients (mean follow-up 26.5 ± 11.7 months) with well-controlled uveitis [52].

Strengths of this study include a large sample size compared to previous studies, the evaluation of standardized documentation of all relevant parameters according to SUN classification and MIWGUC recommendations (comparable to a prospective study case report form), and a long follow-up period to assess ME onset. However, study limitations include the retrospective study design and a potential selection bias at a tertiary center, where the most severe cases are treated.

## 5. Conclusions

In summary, this large-cohort study confirms that ME is a relevant secondary complication in children with JIA-U. Hence, ME may be detected already during the initial stage of severe inflammatory disease and may also develop during the later course of disease. The results of this study contribute to our knowledge of potential risk factors. Ocular findings at first presentation as well as significant changes in certain parameters (e.g., IOP, LFM, and BCVA) during follow-up may help to identify patients at risk for ME onset, indicating a more aggressive anti-inflammatory treatment approach. As ME is a common complication in JIA-U, routine OCT imaging as an objective measurement is recommended in these children.

## Figures and Tables

**Figure 1 jcm-10-04513-f001:**
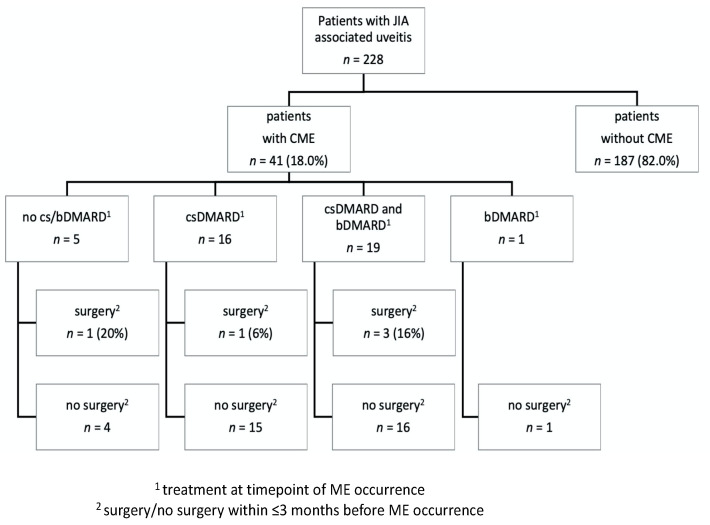
Onset of macular edema (ME) in patients with juvenile idiopathic arthritis-associated uveitis with/without ocular surgery and with/without conventional synthetic (cs) or biological (b) disease-modifying anti-rheumatic (DMARD) treatment.

**Figure 2 jcm-10-04513-f002:**
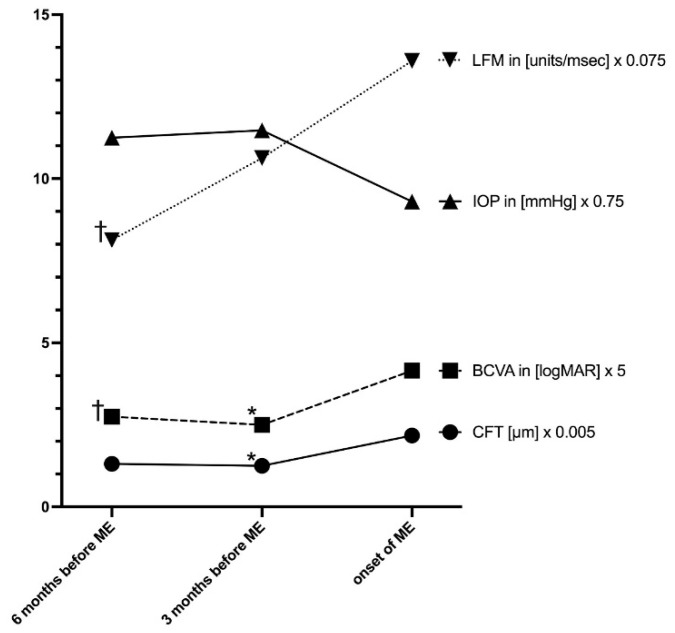
Central foveal thickness (CFT), laser-flare photometry values, best-corrected visual acuity (BCVA), and intraocular pressure (IOP) in patients with juvenile idiopathic arthritis-associated uveitis at 6 months, 3 months before, and at onset of macular edema. Adjusted *p*-value (posthoc analysis for significant parameter) 3 months before ME onset vs. ME onset (* *p* < 0.05); 6 months before ME onset vs. ME onset († *p* < 0.05).

**Table 1 jcm-10-04513-t001:** Macular edema (ME) in patients with juvenile idiopathic arthritis-associated uveitis (JIA-U): baseline characteristics of all study patients.

	JIA-U Total Group	JIA-U w/o ME	JIA-U with ME **	*p*-Value ^a^ w/o vs. with ME
Patients, *n* (%)	228 (100.0)	187 (82.0)	41 (18.0)	n/a
Female gender, *n* (%)	157 (68.9)	123 (65.8)	34 (82.9)	<0.05 ^(1)^
Age at BL, mean ± SD	7.6 ± 4.3	7.1 ± 3.7	10.3 ± 5.5	<0.001
Age at uveitis onset, mean ± SD	5.2 ± 3.1	5.2 ± 2.9	5.6 ± 3.9	0.42
Age at JIA onset, mean ± SD	4.0 ± 3.0	3.8 ± 2.7	4.7 ± 4.1	0.10
Duration of uveitis at BL, mean ± SD	2.4 ± 3.6	1.9 ± 2.9	4.7 ± 5.3	<0.01
JIA subtype, *n* (%)				0.27 ^(2)^
Systemic arthritis	0 (0.0)	0 (0.0)	0 (0.0)	n/a ^(1)^
Polyarthritis RF negative	39 (17.1)	35 (18.7)	4 (9.8)	0.25 ^(2)^
Polyarthritis RF positive	1 (0.4)	1 (0.5)	0 (0.0)	1.00 ^(2)^
Oligoarthritis	173 (75.9)	141 (75.4)	32 (78.0)	0.07 ^(1)^
Enthesitis-related arthritis	3 (1.3)	2 (1.1)	1 (2.4)	0.45 ^(2)^
Psoriatic arthritis	2 (0.9)	1 (0.5)	1 (2.4)	0.33 ^(2)^
Undifferentiated arthritis	10 (4.4)	7 (3.7)	3 (7.3)	0.39 ^(2)^
HLA-B27 positivity, *n* (%)	25 (11.0)	20 (10.7)	5 (12.2)	0.78 ^(2)^
ANA positivity, *n* (%)	211 (94.2)	175 (95.1)	36 (90.0)	0.26 ^(1)^
Uveitis-related eye complications, *n* (%)				
Band keratopathy	61 (26.8)	42 (22.5)	19 (46.3)	<0.01 ^(1)^
Posterior synechiae	73 (32.0)	51 (27.3)	22 (53.7)	<0.01 ^(1)^
Iris rubeosis	10 (4.4)	7 (3.7)	3 (7.3)	0.39 ^(2)^
Cataract	72 (31.6)	46 (24.6)	26 (63.4)	<0.0001 ^(1)^
Optic disc swelling	45 (19.7)	31 (16.6)	14 (34.1)	0.02 ^(1)^
Glaucoma	20 (8.8)	16 (8.6)	4 (9.8)	0.76 ^(2)^
Epiretinal membrane	18 (7.9)	8 (4.3)	10 (24.4)	<0.0001 ^(2)^
Previous eye surgery, *n* (%)				
Cataract surgery	13 (5.7)	10 (5.3)	3 (7.3)	0.70 ^(2)^
Glaucoma surgery	1 (0.4)	0 (0.0)	1 (2.4)	0.17 ^(2)^
Other surgery	7 (3.1)	4 (2.1)	3 (7.3)	0.10 ^(2)^

^a^ For nominal variables, the quantity and the percentage are given and Chi² test with continuity correction ^(1)^ or Fisher’s exact test ^(2)^ applied. For quantitative variables, mean and standard deviation (SD) are shown, and *t*-test was applied (requirements are fulfilled). BL: baseline visit at institute; ANA: antinuclear antibody; n/a: not applicable. ** including ten cases with ME at baseline.

**Table 2 jcm-10-04513-t002:** Patients with juvenile idiopathic arthritis-associated uveitis (JIA-U): occurrence of macular edema (ME). Characteristics at baseline.

Parameter	JIA-U w/o ME (*n* = 187)	JIA-U with ME (*n* = 41)	*p*-Value	Univariable Cox Regression Analysis ^a^		
				HR	95% CI	*p*-Value
Female sex, *n* (%)	123 (65.8)	34 (82.9)	0.05 ^(1)^	1.91	[0.84, 4.32]	0.12
Bilateral uveitis, *n* (%)	134 (71.7)	32 (78.0)	0.52 ^(1)^	1.17	[0.56, 2.44]	0.69
Oligoarthritis, *n* (%)	141 (75.4)	32 (78.0)	0.87 ^(1)^	0.88	[0.42, 1.85]	0.73
ANA positivity,*n* (%)	175 (95.1)	36 (90.0)	0.26 ^(2)^	0.50	[0.18, 1.42]	0.19
HLA-B27 positivity, *n* (%)	20 (10.7)	5 (12.2)	0.78 ^(2)^	1.01	[0.39, 2.57]	0.99
Age at JIA onset, mean ± SD	3.8 ± 2.7	4.6 ± 4.1	0.10	1.07	[0.98, 1.19]	0.15
Age at uveitis onset, mean ± SD	5.2 ± 2.9	5.6 ± 3.9	0.42	1.06	[0.96, 1.16]	0.27
Age at BL, mean ± SD	7.1 ± 3.7	10.2 ± 5.5	<0.001	1.19	[1.11, 1.27]	<0.0001
Uveitis duration at BL, mean ± SD	3.4 ± 3.6	5.1 ± 5.6	<0.01	1.17	[1.10, 1.20]	<0.0001
BCVA in logMAR, mean ± SD	0.3 ± 0.5	0.8 ± 0.7	<0.001	2.49	[1.70, 3.64]	<0.0001
IOP, mean ± SD	14.8 ± 5.5	12.4 ± 4.7	0.17 ^(3)^	0.88	[0.81, 0.96]	<0.01
LFM ^b^ in pu/ms mean ± SD	19.7 ± 38.1	64.2 ± 134.4	0.01 ^(3)^	1.003	[1.00, 1.01]	0.01
ACC grade ^c^ ≥0.5 +, *n* (%)	106 (58.2)	28 (70.0)	0.37 ^(1)^	1.39	[0.72, 2.69]	0.33
Band keratopathy, *n* (%)	42 (22.5)	19 (46.3)	<0.01 ^(1)^	2.29	[1.23, 4.23]	<0.01
Posterior synechiae, *n* (%)	51 (27.3)	22 (53.7)	<0.01 ^(1)^	2.55	[1.38, 4.71]	<0.01
Iris rubeosis, *n* (%)	2 (1.1)	0 (0.0)	0.39 ^(2)^	2.17	[0.67, 7.06]	0.19
Epiretinal membrane, *n* (%)	8 (4.3)	10 (24.4)	<0.0001 ^(2)^	6.19	[2.92, 13.12]	<0.0001
Optic disc swelling, *n* (%)	31 (16.6)	14 (34.1)	0.02 ^(1)^	2.81	[1.46, 5.39]	<0.01
Cataract, *n* (%)	46 (24.6)	26 (63.4)	<0.0001 ^(1)^	4.24	[2.17, 8.26]	<0.0001
Previous cataract surgery, *n* (%)	10 (5.3)	3 (7.3)	0.70 ^(2)^	1.17	[0.36, 3.84]	0.79
Systemic corticosteroids, *n* (%)	39 (20.9)	14 (34.1)	0.07 ^(1)^	1.75	[0.91, 3.37]	0.10
csDMARDs, *n* (%)	147 (79)	35 (89.7)	0.19 ^(1)^	2.26	[0.80, 6.37]	0.12
bDMARDs, *n* (%)	29 (15.6)	6 (15.4)	1.00 ^(1)^	1.44	[0.60, 3.47]	0.41

For nominal variables, the quantity and the percentage are given and Chi² test with continuity correction ^(1)^ or Fisher’s exact test ^(2)^ applied. For quantitative variables, mean and standard deviation (SD) are shown, and *t*-test was applied when requirements were fulfilled; otherwise ^(3)^ a Mann–Whitney-U test was performed. ^a^ Risk of occurrence of ME was calculated with univariable COX regression analysis, and corresponding hazard risks (HR), 95% confidence intervals (CI), and *p*-values are shown. ^b^ Kowa laser-flare device. ^c^ By slit-lamp; *p* < 0.05 statistically significant. JIA: juvenile idiopathic arthritis; BCVA: best-corrected visual acuity; IOP: intraocular pressure; LFM: laser-flare photometry; ACC: anterior chamber cell with reference to SUN (Jabs et al. 2005); csDMARDs: conventional synthetic disease-modifying anti-rheumatic drugs; bDMARDs: biological disease-modifying anti-rheumatic drugs.

**Table 3 jcm-10-04513-t003:** Patients with juvenile idiopathic arthritis-associated uveitis: ocular findings before and at onset of macular edema (ME).

	A: 6 Months before Onset of ME *n* = 29	B: 3 Months before Onset of ME *n* = 23	C: At Onset of ME *n* = 41	*p*-Value (for All)	Adjusted *p*-Value (Posthoc Analysis for Significant Parameter) A vs. B//B vs. C//A vs. C
BCVA logMAR, mean (SD)	0.55 ± 0.51	0.50 ± 0.41	0.83 ± 0.56	0.02 ^(4)^	1.00//0.03//0.03
LFM, pu/ms, mean (SD)	108.4 ± 113.9	141.7 ± 202.6	181.2 ± 183.8	<0.05 ^(4)^	1.00//0.12//0.03
AC cells ≥ 0.5+, *n* (%)	17 (58.6)	12 (52.2)	27 (65.9)	0.56 ^(1)^	
Posterior synechiae, *n* (%)	8 (27,6%)	16 (70.0%)	17 (41.5%)	0.33 ^(1)^	
Band keratopathy, *n* (%)	17 (58.6)	14 (63.6)	23 (56.1)	0.93 ^(1)^	
Cataract, *n* (%)	24 (82.8)	19 (82.6)	32 (78.0)	0.89 ^(2)^	
Iris rubeosis, *n* (%)	0 (0.0)	0 (0.0)	1 (2.4)	1.00 ^(2)^	
Optic disc swelling, *n* (%)	4 (13.8)	5 (21.7)	20 (48.8)	<0.01 ^(1)^	1.00//0.18//0.02
IOP mmHg, mean (SD)	15.0 ± 5.4	15.3 ± 4.1	12.4 ± 4.8	0.04 ^(3)^	0.98//0.09//0.09
Epiretinal membrane formation, *n* (%)	4 (13.8)	5 (21.7)	15 (36.6)	0.09 ^(1)^	
CFT µm, mean (SD)	262 ± 48	250 ± 48	435 ± 193	<0.01 ^(4)^	1.00//0.01//>0.05
csDMARD use, *n* (%)	25 (86.2)	19 (82.6)	36 (87.8)	1.00 ^(2)^	
Methotrexate use, *n* (%)	21 (72.4)	15 (65.2)	28 (68.3)	0.85 ^(1)^	
bDMARDs use, *n* (%)	15 (51.7)	13 (56.5)	20 (48.8)	0.84 ^(1)^	
Adalimumab use, *n* (%)	10 (34.5)	9 (39.1)	16 (38.4)	0.91 ^(1)^	
Systemic corticosteroids > 10 mg/day; *n* (%)	0 (0.0)	3 (13.0)	3 (7.3)	0.12 ^(2)^	
Topical corticosteroids >2 daily doses, *n* (%)	14 (48.3)	8 (34.8)	24 (58.5)	0.19 ^(1)^	
Systemic CA inhibitors, *n* (%)	3 (10.3)	2 (8.7)	3 (7.3)	0.91 ^(2)^	
Systemic NSAIDs, *n* (%)	9 (31.0)	7 (30.4)	13 (31.7)	0.99 ^(1)^	
Topical NSAIDs, *n* (%)	0 (0.0)	0 (0.0)	2 (4.9)	0.48 ^(2)^	

For nominal variables, the quantity and the percentage are given and Chi² test with continuity correction ^(1)^ or Fisher’s exact test ^(2)^ applied. For quantitative variables, mean and standard deviation (SD) are shown, and ^(3)^ ANOVA was applied if requirements were fulfilled, followed by a Tukey’s posthoc analysis; otherwise, we performed ^(4)^ a Kruskal–Wallis test followed by Dunn’s posthoc analysis. BCVA, best-corrected visual acuity; AC, anterior chamber; LFM, laser-flare photometry; IOP, intraocular pressure; CFT, central foveal thickness; DMARD, disease-modifying anti-rheumatic drug; CA, carboanhydrase; NSAIDs, nonsteroidal anti-inflammatory drugs.

## Data Availability

The data presented in this study are available on request from the corresponding author.

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
