# Peer review of "Occurrence and Risk Factors for Macular Edema in Patients with Juvenile Idiopathic Arthritis-Associated Uveitis"

_jcm, 2021, doi:10.3390/jcm10194513_

Round 1
Reviewer 1 Report
This a very nice article, easy to read and nicely written.
I can give my appreciation as an internist treating eye inflammation : gives interesting information about the risk of macular oedema in this patient population.
If it was possible, I had a question about the biologic treatments used : apparently, a few patients receive biologic Dmards, but not adalimumab. What was then the bDmard used ?
No other remarks.
Author Response
Reviewer #1
This a very nice article, easy to read and nicely written.
I can give my appreciation as an internist treating eye inflammation: gives interesting information about the risk of macular oedema in this patient population.
If it was possible, I had a question about the biologic treatments used: apparently, a few patients receive biologic Dmards, but not adalimumab. What was then the bDmard used?
No other remarks.
Authors’ response: The following sentence has now been added in the results section: “At timepoint of ME occurrence, patients were treated with adalimumab (n=16), infliximab (n=2), abatacept (n=1) and golimumab (n=1).” In table 3 we have listed only adalimumab, as the statistical power for the analysis of the other bDMARDs was too low.
Reviewer 2 Report
The authors demonstrate clearly the risk factors for macular edema (ME) in juvenile idiopathic arthritis-associated uveitis (JIA-U).  The data and methods may certainly be of use for planning treatment strategies in JIA-U.
As indicated by the authors, epiretinal membrem (ERM) may be due to previous severe inflammation, but we cannot completely deny the possibility that structural changes due to ERM may have caused yellow team edema. Although this may be difficult due to the small number of JIA-U patients with yellow squad edema, have you considered the risk factor for ME in JIA-U patients without ERM?
Authors wrote <Although all our patients were managed according to international treatment recommendations using a step-ladder approach from topical corticosteroids, to csDMARDs, and to bDMARds, ME onset could still not be prevented in patients at risk.> in discussion. The percentage of JIA-U with bDMARds has decreased between 3 months before and at the onset of ME. In other words, the above rationale needs to be explained in detail.
Author Response
Reviewer #2
The authors demonstrate clearly the risk factors for macular edema (ME) in juvenile idiopathic arthritis-associated uveitis (JIA-U). The data and methods may certainly be of use for planning treatment strategies in JIA-U.
As indicated by the authors, epiretinal membrem (ERM) may be due to previous severe inflammation, but we cannot completely deny the possibility that structural changes due to ERM may have caused yellow team edema. Although this may be difficult due to the small number of JIA-U patients with yellow squad edema, have you considered the risk factor for ME in JIA-U patients without ERM?
Authors’ response: We are not sure, what the reviewer means with “yellow team edema”; we assume that this probably means “macular edema”: ERM was highly significant in the univariable Cox regression analysis but was not an independent risk factor for ME onset in the multivariable analysis, as we have mentioned in the results section. This highlights the fact, that ERM is not an independent risk factor but is a structural complication that may occur in JIA-uveitis children together with ME – mostly after severe or long-lasting inflammation. However, ME often occurred in the absence of ERM. Our analysis enclosed enumerated other significant risk factors for ME.
Authors wrote <Although all our patients were managed according to international treatment recommendations using a step-ladder approach from topical corticosteroids, to csDMARDs, and to bDMARds, ME onset could still not be prevented in patients at risk.> in discussion. The percentage of JIA-U with bDMARds has decreased between 3 months before and at the onset of ME. In other words, the above rationale needs to be explained in detail.
Authors’ response: The number of patients with bDMARDs did not change significantly between timepoints -6 vs. -3 and -3 vs. 0 (see Table 3). In selected patients, bDMARDs had to be stopped (e.g., by adverse events, and interruption of health insurance cost approval).
The following sentence has now been added to the results section: “Compared to 3 months before, the percentage of patients with bDMARDs at timepoint of ME onset slightly but not significantly decreased (Table 3), as bDMARDs had to be stopped due to adverse events or interruption of health insurance cost approval.”